# A novel CSP C-terminal epitope targeted by an antibody with protective activity against *Plasmodium falciparum*

Nathan Beutler[1]☯, Tossapol Pholcharee[2]☯, David Oyen[2]☯¤a, Yevel Flores-Garcia[3], Randall S. MacGill[4], Elijah Garcia[1], Jaeson Calla[5], Mara Parren[1], Linlin Yang[1]¤b, Wayne Volkmuth[6]¤c, Emily Locke[4], Jason A. Regules[7], Sheetij Dutta[7], Daniel Emerling[6], Angela M. Early[8,9], Daniel E. Neafsey[8,9], Elizabeth A. Winzeler[5], C. Richter King[4], Fidel Zavala[3], Dennis R. Burton[1,10]*, Ian A. Wilson[2,11]*, Thomas F. Rogers[1,12]*

1 Department of Immunology and Microbiology, The Scripps Research Institute, La Jolla, California, United States of America, 2 Department of Integrative Structural and Computational Biology, The Scripps Research Institute, La Jolla, California, United States of America, 3 Malaria Research Institute, Johns Hopkins Bloomberg School of Public Health, Baltimore, Maryland, United States of America, 4 PATH's Malaria Vaccine Initiative, Washington, District of Columbia, United States of America, 5 Department of Pediatrics, University of California, San Diego, School of Medicine, La Jolla, California, United States of America, 6 Atreca Inc., South San Francisco, California, United States of America, 7 Malaria Biologics Branch, Walter Reed Army Institute of Research, Silver Spring, Maryland, United States of America, 8 Infectious Disease and Microbiome Program, Broad Institute of MIT and Harvard, Cambridge, Massachusetts, United States of America, 9 Department of Immunology and Infectious Diseases, Harvard T.H. Chan School of Public Health, Boston, Massachusetts, United States of America, 10 Ragon Institute of Massachusetts General Hospital, Massachusetts Institute of Technology, and Harvard University, Cambridge, Massachusetts, United States of America, 11 The Skaggs Institute for Chemical Biology, The Scripps Research Institute, La Jolla, California, United States of America, 12 Division of Infectious Diseases, Department of Medicine, University of California, San Diego, La Jolla, California, United States of America

☯ These authors contributed equally to this work.
¤a Current address: Pfizer Inc., San Diego, California, United States of America.
¤b Current address: Perelman School of Medicine, University of Pennsylvania, Philadelphia, Pennsylvania, United States of America.
¤c Current address: Bluestar Genomics, San Mateo, California, United States of America.
* burton@scripps.edu (DRB); wilson@scripps.edu (IAW); trogers@scripps.edu (TFR)

**Data Availability Statement:** The crystal structures of all Fab-peptide complexes have been deposited in the Protein Data Bank with accession codes:

## Abstract

Potent and durable vaccine responses will be required for control of malaria caused by *Plasmodium falciparum (Pf)*. RTS,S/AS01 is the first, and to date, the only vaccine that has demonstrated significant reduction of clinical and severe malaria in endemic cohorts in Phase 3 trials. Although the vaccine is protective, efficacy declines over time with kinetics paralleling the decline in antibody responses to the *Pf* circumsporozoite protein (*Pf*CSP). Although most attention has focused on antibodies to repeat motifs on *Pf*CSP, antibodies to other regions may play a role in protection. Here, we expressed and characterized seven monoclonal antibodies to the C-terminal domain of CSP (ctCSP) from volunteers immunized with RTS,S/AS01. Competition and crystal structure studies indicated that the antibodies target two different sites on opposite faces of ctCSP. One site contains a polymorphic region (denoted α-ctCSP) and has been previously characterized, whereas the second is a previously undescribed site on the conserved β-sheet face of the ctCSP (denoted β-ctCSP). Antibodies to the β-ctCSP site exhibited broad reactivity with a diverse panel of ctCSP peptides

7RXI, 7RXJ, 7S0X, 7RXL, and 7RXP for Fab234, 236, 352, 1488, and 1512 in complex with ctCSP, respectively.

**Funding:** This work was supported by PATH's Malaria Vaccine Initiative (IAW), the Bill & Melinda Gates Foundation (grant no. OPP1170236 (IAW, DRB) and INV-004923 (IAW,DRB,TFR)) under collaborative agreements with The Scripps Research Institute, U.S. Agency for International Development (USAID) Innovations in Malaria Vaccine Development Contract Number 7200AA20C00017 (CRK,IAW) and Bloomberg Philanthropies (YFG,FZ). The funders did not play any role in the study design, data collection and analysis, decision to publish, or preparation of the manuscript.

**Competing interests:** We have read the journal's policy and the authors of this manuscript have the following competing interests. The authors N.B., T. P., D.O., Y.F.G., R.S.M., E.G., J.C., M.P., L.Y., E.L., J.A.R., S.D., A.M.E., D.E.N., E.W., C.R.K., F.Z., D.R. B., I.A.W., and T.F.R. declare that they have no competing interests. W.V., D.E. were or are employees of Atreca, Inc. and own equity in Atreca, Inc.

whose sequences were derived from field isolates of *P. falciparum* whereas antibodies to the α-ctCSP site showed very limited cross reactivity. Importantly, an antibody to the β-site demonstrated inhibition activity against malaria infection in a murine model. This study identifies a previously unidentified conserved epitope on CSP that could be targeted by prophylactic antibodies and exploited in structure-based vaccine design.

## Author summary

The most advanced malaria vaccine candidate to date, RTS,S, is composed of the central repeat region, so called because it consists of repeats of an NANP amino-acid sequence, and the C-terminal domain from the *Plasmodium falciparum* circumsporozoite protein (*Pf*CSP). RTS,S is about 50% effective against the liver stage of the malaria parasite, but its efficacy decreases over time, concomitant with waning of antibodies that target *Pf*CSP. Thus, further understanding of which antibodies are effective in the immune response to *Pf*CSP is needed to facilitate design of next-generation malaria vaccines. While much is known about antibodies to the NANP repeat region, the nature and efficacy of antibodies that target the *Pf*CSP C-terminal domain (ctCSP) is underexplored. Here, we characterize antibodies against ctCSP that were derived from volunteers in a phase 2a trial of RTS,S with a fractional dose regimen. We find that some antibodies bind to a previously identified polymorphic site on ctCSP, but others bind to a novel site that is highly conserved across different *P. falciparum* isolates. Furthermore, these antibodies show protection against *P. falciparum* infection in a mouse model. Thus, a previously unidentified and conserved site on ctCSP can be targeted by antibodies and will aid in design of more effective next-generation *Pf*CSP-based malaria vaccines and therapeutics.

## Introduction

Malaria continues to be a major global health priority with an estimated 229 million cases and 409,000 deaths in 2019 [1]. Increased resistance to antimalarial drugs has heightened concerns as widely used drugs like artemisinin, which is commonly used to combat chloroquine resistance, is increasingly failing [2,3]. The spread of drug resistance in regions of Africa is of particular concern since this continent accounted for 94% of estimated malaria cases and 94% of estimated deaths from malaria in 2019 [1]. To avoid the continued spread of resistance to antimalarial drugs and as a tool to limit overall malaria disease and death, the development of an effective vaccine against malaria has been prioritized [4,5]. Recent vaccine candidates target *Plasmodium falciparum*, which accounted for an overwhelming majority of reported malaria infections in the WHO African Region in 2019 [1,6,7]. Currently, RTS,S/AS01 is the most advanced vaccine against malaria and has been recently approved by WHO for use in children in Africa. The vaccine is composed of a virus-like particle containing 19 NANP repeats and the C-terminal region of the strain 3D7 *Pf*CSP linked to the hepatitis B surface antigen protein (HBsAg), and also includes unmodified HBsAg along with adjuvant AS01E [8]. The vaccine showed an efficacy of approximately 50% within the first 14 months of administration in phase 3 clinical trials in Africa. However, its efficacy decreased over time [6,9] and this decrease was attributed to waning titers of anti-CSP antibodies [10]. A related vaccine candidate, R21, is composed of the same HBsAg-CSP fusion without unmodified HBsAg and formulated with Matrix-M adjuvant [11]. In a recent phase 2 clinical trial, R21 was administered

to children in Burkina Faso, with a 3-dose regimen at 4-week intervals prior to the malaria season and a 4th dose one year later. Efficacy levels of 74%, and 77%, in the low- and high-dose adjuvant groups respectively, were observed in a 12-month follow-up period following dose 3 [12]. However, only a single episode of malaria was reported in the control group after day 200, suggesting that essentially all cases were averted during the first ~6 months of follow-up [13]. As with RTS,S/AS01 immunization, the R21/Matrix-M-induced anti-NANP antibody titers also quickly waned over time to almost baseline levels prior to the 4th dose [12].

*Pf*CSP has multiple domains and regions that include the N-terminal domain, central NANP repeat region [14] and ctCSP, which is largely composed of an α-thrombospondin repeat (αTSR) domain [15] that has been suggested to interact with heparan sulfate proteoglycans in the process of liver cell invasion [16,17].The so-called junctional region is located between the N-terminal domain and central repeat region; it contains an NPDP sequence and three NVDP motifs within a minor repeat region, where both motifs are related to the dominant NANP motif [18–22]. Monoclonal antibodies (mAbs) to the central repeat region and to the junctional region have been shown to provide a protective effect in a mouse model of malarial disease [18,21,23–25]. Furthermore, in passive immunization studies in humans, a mAb to the junctional region has been shown to offer protection against controlled human malaria challenge [26]. Vaccine immunogenicity studies have shown the NANP repeat region to be the immunodominant B-cell epitope [10]. Despite this immunodominance, C-terminal antibodies can also be elicited and have been associated with vaccine efficacy in humans [27–30]. Of further note, genetic variability has been observed in ctCSP of *P. falciparum* [31,32]. Given the partial efficacy of the RTS,S vaccine, full understanding of the spectrum of epitopes in the CSP antigens that are recognized by protective antibodies and the role that they play in protection may contribute to design of improved next-generation *Pf*CSP vaccines. While potent antibodies against the NANP repeats [19,23–25,33–35], minor repeats [22], and junctional region [18–21] have been extensively characterized, the roles of *Pf*CSP C-terminal and N-terminal antibodies remain understudied.

Here, we aimed to characterize and define mAbs that have specificity to epitopes located on ctCSP. The antibodies were isolated from volunteers immunized with the RTS,S/AS01 vaccine in a phase 2a trial that compared three standard doses (50 μg) to two standard doses and a fractional third dose (10 μg) [36]. We selected seven C-terminal specific mAbs with divergent sequences and different clonotypes. We then grouped them into distinct epitope bins by competition assay [37], and characterized them for affinity and breadth to further elucidate their potential role in protection against *P. falciparum*. We obtained crystal structures of the Fab fragments for five of these antibodies in complex with ctCSP, one of which revealed a new epitope region consisting of one face of ctCSP comprised of β-strands in the conserved αTSR homology region. We further showed that one member of this new class of anti-ctCSP mAbs had anti-parasite activity by conducting an in vivo protection study in mice using transgenic *P. berghei* that express PfCSP, although further studies are required to generalize this finding. Overall, this study reveals a conserved ctCSP epitope region that binds antibodies with wide breadth, high affinity, and protective ability, which could be utilized in the design of antibodies as medical countermeasures and next-generation malaria vaccines.

## Results

### Competition experiments reveal two classes of ctCSP antibodies

The antibodies analyzed in this study (Fig 1A) were derived from volunteers in a phase 2a clinical trial that compared standard dosing of the RTS,S/AS01 vaccine with a delayed fractional dose regimen [36]. All vaccinated subjects in this trial showed positive titers for *Pf*CSP,

**A**

| mAb | Group | Heavy chain V gene | CDR H3 (Kabat) | Light chain V gene | CDR L3 (Kabat) | H/L SHM (a.a.)* |
|---|---|---|---|---|---|---|
| 234 | Fx017M | IGHV3-21*01, or IGHV3-21*02, or IGHV3-21*03 | ARGFIQFHYYMDV | IGLV3-21*03, or IGLV3-21*02, or IGLV3-21*01 | QVWHSSSDPVV | 8/6 |
| 236 | Fx017M | IGHV3-48*01, or IGHV3-48*04, or IGHV3-21*05 | GGSLHPSAGADWIDP | IGLV3-1*01, or IGLV3-21*02, or IGLV3-21*01 | QAWDSNTYV | 9/10 |
| 352 | Fx017M | IGHV3-21*05, or IGHV3-21*01, or IGHV3-21*02 | GMGIAVRRFDY | IGLV3-21*02, or IGLV3-21*03, or IGLV3-21*01 | QVWDSSTVAS | 11/7 |
| 1488 | 012M | IGHV3-21*01, or IGHV3-21*02, or IGHV3-21*03 | RAGGFDAYYFDY | IGLV3-1*01, or IGLV3-25*02, or IGLV3-25*03 | QAWDSSTVV | 5/2 |
| 1504 | 012M | IGHV4-39*01, or IGHV4-39*01, or IGHV4-39*01 | LGVRWELLVGGFVNNWFDP | IGKV3-15*01, IGKV3D-15*01, IGKV3D-15*03 | QQYNNWLGT | 5/2 |
| 1512 | Fx017M | IGHV3-21*01, or IGHV3-21*04, or IGHV3-21*07 | DDNILGHCRSKSCQRNYYHGMDV | IGKV2-28*01, or IGKV2D-28*01, or IGKV2D-28*02 | MQALQTPFT | 11/3 |
| 1550 | 012M | IGHV3-23*01, or IGHV3-23D*01, or IGHV3-23*04 | DVSYFDSPGYYYFDY | IGKV1-33*01, IGKV1D-33*01, IGKV1D-13*01 | QQYDNLPLT | 7/4 |
| 1710 (Scally et al., 2017) | N/A | IGHV3-21*01, or IGHV3-21*02, or IGHV3-21*03 | DPGIAAADNHWFDP | IGLV3-1*01, or IGLV3-25*02, or IGLV3-25*03 | QAWDSSTVV | 2/3 |

**B**

**Competing Abs**

| Saturating Abs | 236 | 234 | 352 | 1488 | 1512 | 1504 | 1550 |
|---|---|---|---|---|---|---|---|
| 236 | 101.9 | 74.3 | 90.5 | 99.5 | 87.4 | 89.8 | 102.9 |
| 234 | 103.1 | 99.1 | 101.3 | 104.5 | 93.8 | 100.3 | 103.9 |
| 352 | 102 | 85.6 | 99.6 | 100.2 | 92.6 | 99.2 | 105.4 |
| 1488 | 95.3 | 74.2 | 87.6 | 95 | 93.8 | 99.9 | 110.1 |
| 1512 | 11.9 | -7.4 | -20.1 | -19.7 | 97.3 | 79.2 | 98.4 |
| 1504 | -6.5 | 24 | 13.7 | 11.1 | 92.7 | 94.5 | 94.9 |
| 1550 | 5.5 | 9.6 | 6.4 | 8.6 | 90.5 | 65 | 99.3 |

Non-competing pairs   Self-competing pairs   Competing pairs

**Fig 1. In vitro binning of ctCSP specific mAbs against recombinant CSP.** (**A**) Immunogenetics ctCSP-specific mAbs. mAbs were isolated in a previous study (36) from donors undergoing both standard vaccination (months 0, 1, 2; 012M) and delayed fractional dosing (full dose at months 0, 1 + 1/5 dose at month 7; Fx017M). All mAbs were isolated from unique donors, except mAbs 234 + 236, and 1504 + 1550 were each derived from the same individual. The germline genes were identified using IgBLAST (NIH) with the three top hits reported, and the heavy/light chain somatic hypermutations (H/L SHM) of amino acid residues were derived from the comparison with the first top-hit gene. The complementarity-determining regions (CDRs) were defined based on the Kabat system. (**B**) Binning experiments were performed using bio-layer interferometry on an Octet HTX at 25˚C. An in-tandem binning assay format was set up. Biotinylated *Pf*CSP was loaded onto streptavidin sensors. Loaded sensors were dipped into saturating Ab followed by competing Ab. Values are displayed as percent inhibition of binding to *Pf*CSP (see Materials and Methods). Pairs with greater than 50% inhibition were considered competing pairs.

NANP$_6$, and C-terminal domain (ctCSP) reactive antibodies in sera following vaccination. Sequencing of the variable regions of paired heavy and light chain messenger RNA was conducted from peripheral blood mononuclear cell (PBMC)–sorted plasmablasts sampled 7 days after the third immunization [38]. Results of the somatic hypermutation frequency have previously been reported [36] and a full analysis of expressed antibody sequences is being prepared for publication. Sequences for expression of mAbs were selected to sample a variety of IgG

lineages from vaccinee plasmablast responses based on multiple properties, including degree of cellular and/or clonal expansion, level of somatic hypermutation, and sequence convergence among vaccinees. For this study, we did not attempt to evaluate if these antibodies were necessarily characteristic of the level or specificity of the antibodies present in the sera of individual or grouped subjects. Expressed mAbs were screened by ELISA for reactivity to full-length CSP, followed by NANP$_6$, $Pf$16 C-term peptide (residues 283–375 of $P.\ falciparum$ 3D7 CSP), and HBsAg as previously described [36]. From 369 antibodies screened, 20 bound to ctCSP. We structurally and functionally characterized some of the other antibodies to the NANP repeats from this vaccine trial in previous studies [24,25,33,34]. The sequences studied here are specifically reactive to $Pf$16 C-term peptide in ELISA and further selected to represent families with different VH and VL gene usage. Competition experiments were performed using bio-layer interferometry (BLI) on an Octet HTX system to define the antibody response to the ctCSP epitope (Fig 1B). The competition assay consisted of capturing biotinylated ctCSP (residues 283–375 of $P.\ falciparum$ 3D7 CSP) onto streptavidin sensors, followed by saturation of the first mAb (mAb1). The ctCSP + mAb1 coated sensors were incubated with a second mAb (mAb2) to monitor competition against similar epitopes (no/low additional binding signal) or to identify different epitopes (high additional binding signal). Epitope competition for each mAb pair was calculated as the percent inhibition of mAb2 binding by the saturated mAb1. Briefly, mAb2 control binding was first evaluated by averaging three non-competitive binding events of mAb2 at equilibrium. Percent inhibition was then calculated as 100 - [(mAb2 binding from competition experiment / mAb2 binding from control binding experiment) x 100] (Fig 1B). The mAb pairs with greater than 50% inhibition were considered as competing pairs that bind to the same region (Fig 1B). When mAbs 236, 234, 352, or 1488 were applied as mAb1, binding of the entire set of tested mAb2 antibodies was inhibited (Fig 1B). However, when the sensors were first coated with mAbs 1512, 1504, or 1550, binding by mAbs 236, 234, 352, and 1488 was not inhibited suggesting some difference in the mode of binding of the two sets of antibodies (Fig 1).

## Crystal structures reveal binding to a previously undescribed epitope region on ctCSP

We determined crystal structures of five Fabs to elucidate how these antibodies recognize their cognate epitopes on the ctCSP or αTSR domain (residues 310–375 of $P.\ falciparum$ 3D7 CSP) (Fig 2 and S1 Table). Fabs 234, 236, 352, and 1488 all recognize the same region on ctCSP, which is similar to the epitope of the previously published mAb1710 [37] (Fig 2A–2C). We refer to this region as the alpha epitope region (α-ctCSP) as it consists of an α-helix that includes the T-cell epitope Th2R (region III), and the so-called CS flap, which contains another T-cell epitope Th3R [15] (Fig 2B). These Th2R and Th3R regions are known to have the highest sequence variation among different $P.\ falciparum$ isolates [39]. The alpha epitope also contains the conserved hydrophobic pocket that is positioned between the CS flap and the α-helix (Fig 2B) and has been hypothesized to bind a hydrophobic ligand on the host cell [15]. The four α-ctCSP-binding Fabs, together with Fab1710 [37], are derived from the $IGHV3$-$21$ germline gene or closely related $IGHV3$-$48$ gene and utilize either the $IGLV3$-$21$ or $IGLV3$-$1$ light chain $V$ gene (Fig 1A). For Fabs 234, 236, 352, 1488, and 1710, the heavy chain buried surface area (BSA) ranges from 401 to 519 Å$^2$ (representing on average 73% total BSA) and contributes more to the binding interaction than the light chain (151 to 187 Å$^2$, on average 27% total BSA) (S2 Table). Complementarity-determining regions (CDRs) H1 and H2 interact with the CS flap and CDRs L1 and L2 interact with the α-helix (Figs 2C and S1). Fabs derived from the same $IGHV$ gene share a similar usage of residues in CDR H2 to interact with the CS flap, whereas conserved usage of some residues in CDRs L1 and L2 is also observed in Fabs that are

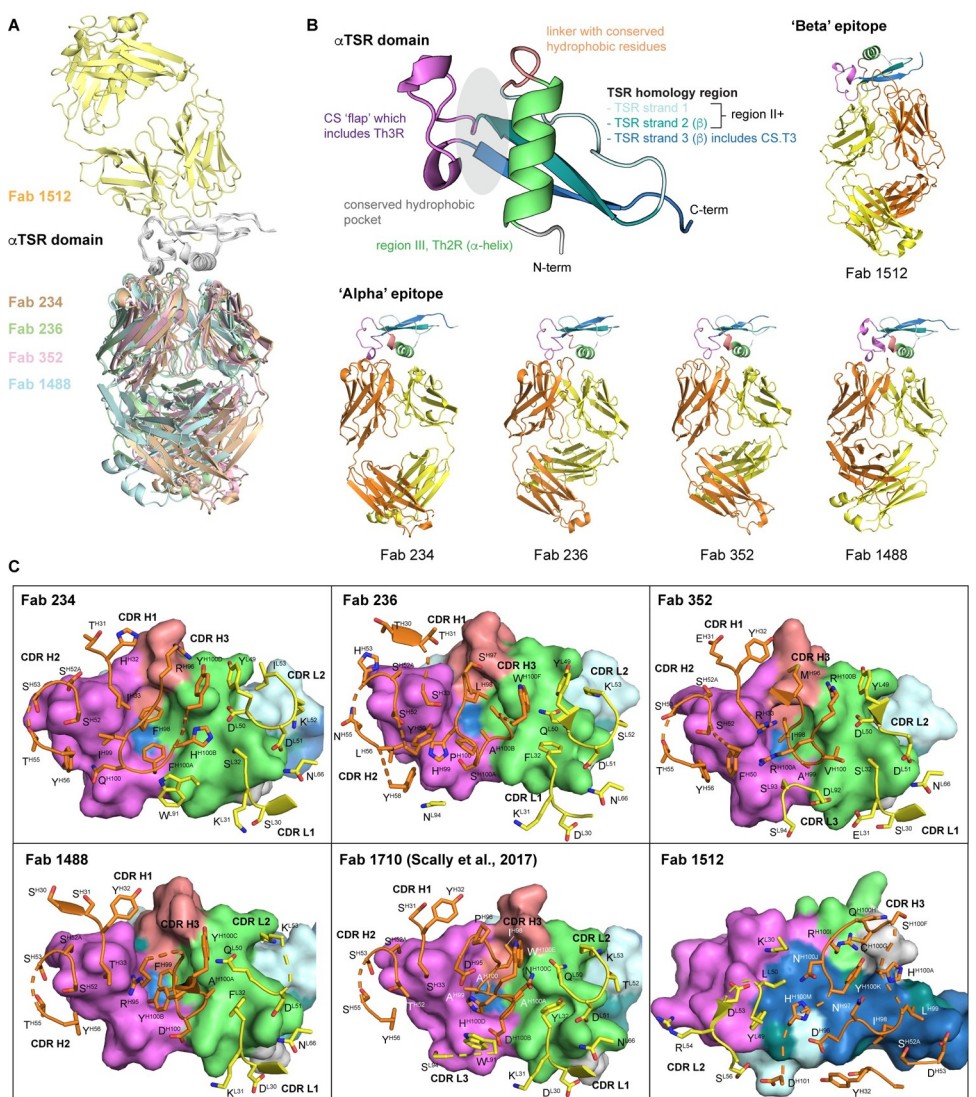

**Fig 2. Crystal structures of anti-ctCSP Fabs in complex with the αTSR domain.** (**A**) Alignment of Fab234, 236, 352, 1488, and 1512 complexes (colored as shown) interacting with the αTSR domain (grey). The complexes are shown in ribbon cartoon representation. (**B**) Ribbon representation of the αTSR domain with important regions labeled and colored. The epitopes on the αTSR domain can be classified into: 1) the α-epitope region recognized by Fabs 234, 236, 352, and 1488; and 2) the β-epitope region recognized by Fab 1512. The Fab heavy chain and light chain are colored orange and yellow, respectively. (**C**) The epitopes of the Fabs are represented as surfaces and colored based on the different regions on the αTSR domain as in (**B**). The paratopes are shown as cartoons with side chains as sticks. The heavy chain and light chain are colored as in (**B**).

encoded by the same *IGLV* gene (Figs 2C and S1). Furthermore, all α-ctCSP-binding Fabs share a similar feature of utilizing CDR H3 to insert into the conserved hydrophobic core between the α-helix and CS flap in the αTSR domain (Fig 2C). These CDR H3s contain hydrophobic residues, such as Phe, Ile, Leu, and Ala, that interact with the hydrophobic core. Interestingly, α-ctCSP-binding antibodies 234, 236, and 352 exhibit a moderate level of somatic hypermutation (SHM) (Fig 1A) with some mutations in residues in CDRs H1 and H2 that contribute to epitope binding (S2 Fig). In comparison, mAb1488 exhibits minimal levels of SHM, similar to the previously reported mAb1710 (S3 Fig). However, this antibody does not

 

make as extensive contacts with the epitope as shown by its smaller binding site (S2 Table), which is consistent with its lower affinity to ctCSP (Fig 3B).

In contrast to the mAbs described above, Fab1512 binds to an epitope that consists of the β-sheet in the TSR homology region that is on the opposite side of the α-ctCSP region and has been termed the beta epitope (β-ctCSP) here (Fig 2A). TSR strands 1 and 2 contain a conserved region II+ and strand 3 includes another T-cell epitope CS.T3 (Fig 2B), which is highly conserved among *P. falciparum* isolates (Fig 3A). Similar to the α-ctCSP Fabs, Fab1512 has a higher contribution from the heavy chain (BSA 548 Å$^2$) than the light chain (150 Å$^2$) for recognition of the αTSR domain (S2 Table), but intriguingly its interactions with the β epitope are mediated predominantly by CDRs L2 and H3 (Fig 2A). Fab1512 CDR L2 interacts with the CS flap, whereas CDR H3 spans the TSR homology strands 1, 2, and 3 and forms extensive hydrogen-bonding networks, which are mostly water-mediated, with the β-ctCSP epitope (S4 Fig). Furthermore, mAb1512 exhibits a moderate level of SHM in the *IGHV* gene, comparable to mAb 352, and a long, 23-amino-acid, CDR H3, which is not commonly seen in anti-CSP antibodies identified to date [18–25,34,35,37] (Fig 1A). These mutations and long CDR H3 in Fab1512 may have evolved to optimize binding to the β-sheet in the αTSR domain. Overall, the crystal structure of Fab1512 represents the first reported structure of a mAb that recognizes this novel epitope on ctCSP, which exhibits almost no polymorphisms compared to the classical α epitope (Fig 3A) and, therefore, could allow for greater breath of antibody responses to circulating *P. falciparum* strains.

## β-ctCSP specific mAbs display broad binding at high affinity

To further define the binding characteristics of β-ctCSP specific mAbs, we performed surface plasmon resonance (SPR) experiments using recombinant ctCSP peptides. To determine both breadth and affinity, we generated a panel of peptides that correspond to different ctCSP haplotypes as in the previous study [28]. These peptides represent the extensive diversity of ctCSP sequences found naturally in *P. falciparum* strains, including those observed in the Phase 3 ancillary genotyping study, which was conducted to examine differential RTS,S/AS01 vaccine efficacy in the phase III trials [31] (Fig 3A). The laboratory-adapted *P. falciparum* 3D7 strain, which is also the RTS,S sequence source, was used as the reference sequence for the binding experiments (Fig 3A). This panel shows high variability in the T-cell epitopes, Th2R and Th3R, but high conservation in region II+ (RII+) (Fig 3A). Our ctCSP mAb panel contains both α-ctCSP specific mAbs (236, 234, 352, 1488) and β-ctCSP specific mAbs (1512, 1504, and 1550). mAb5D5, a murine derived N-terminal *Pf*CSP specific mAb that was engineered with a human Fc domain [40] was used as a negative control in binding measurements against ctCSP peptides. To measure affinities, IgGs were flowed over CM5 chips that were immobilized with anti-human Fc antibodies and followed by multiple injections of increasing concentrations of ctCSP peptides. Antibodies that exhibited an SPR binding response equal to or less than that of humanized mAb 5D5 were classified as not binding (NB) (Figs 3B and S5). All ctCSP mAbs showed high affinity towards the reference 3D7 ctCSP peptide with dissociation constants (K$_D$) ranging from $6.9 \times 10^{-10}$ M to $2.8 \times 10^{-12}$ M (Figs 3B and S5). Of the α-ctCSP specific mAbs (236, 234, 352, 1488), 234 showed the greatest breadth with binding to 5 of the 15 peptides. 236 showed the highest single affinity of all mAbs towards the H234 haplotype, although like the other α-ctCSP mAbs 352 and 1488, failed to bind to 13 of the 15 peptides (Figs 3B and S5). In contrast, all three β-ctCSP specific mAbs bound to all ctCSP peptides with 1512 displaying the highest average affinity with a K$_D$ of $\sim 10^{-10}$ M (Figs 3B and S5). Overall, β-ctCSP mAbs exhibit greater breadth while maintaining similar affinity as their α-ctCSP counterparts.

 

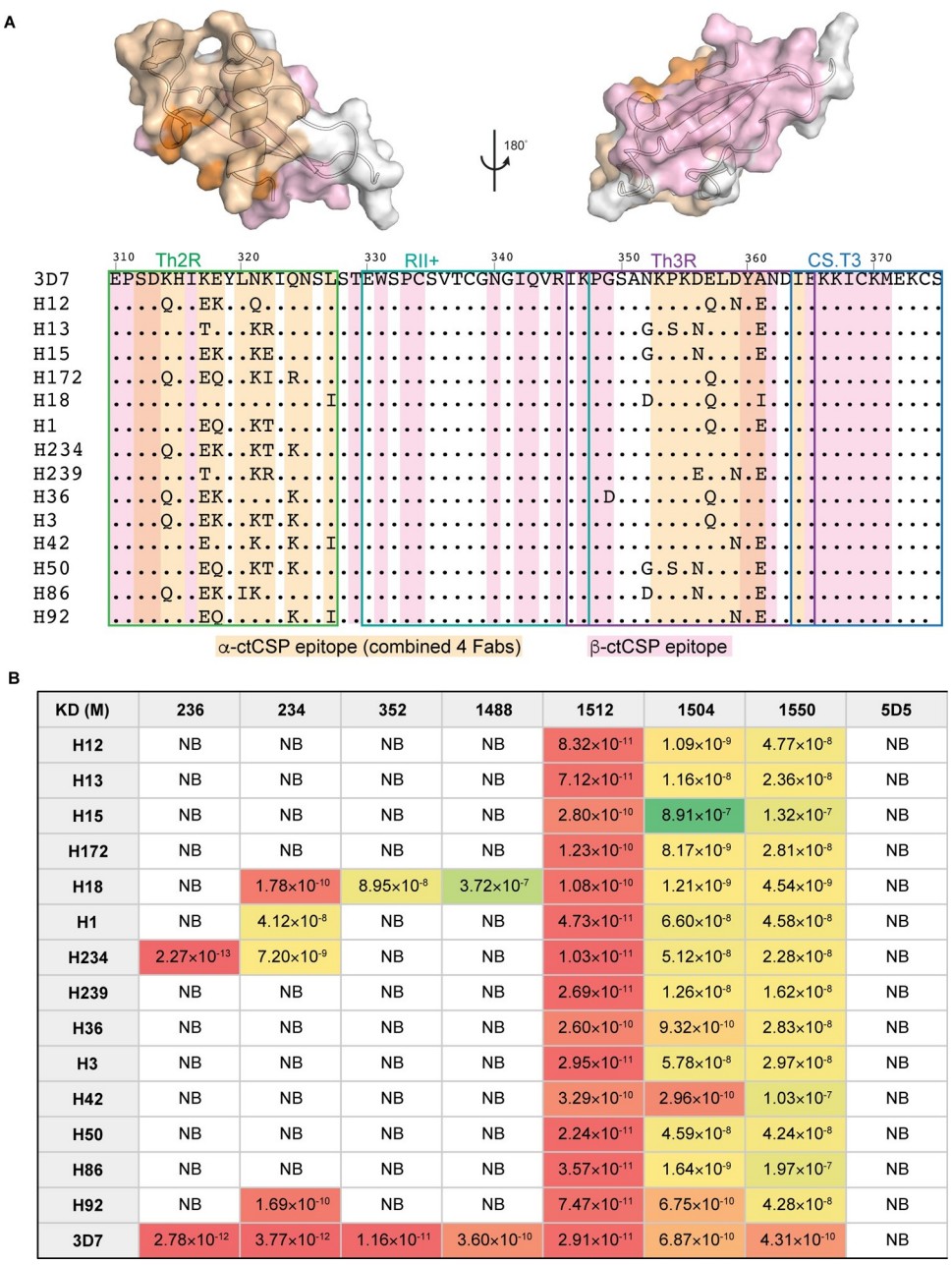

**Fig 3. Sequences of αTSR in different haplotypes of *P. falciparum* and binding of mAbs to peptides corresponding to different haplotypes.** A panel of ctCSP peptides were generated for mAb affinity measurements. (**A**) All ctCSP haplotypes that represent strains found across East and West Africa (31) were referenced against the C-terminal domain of PfCSP isolate 3D7. The Th2R, RII+, Th3R, and CS.T3 regions are enclosed in green, cyan, magenta, and blue boxes respectively. The combined α-ctCSP epitopes from mAbs 234, 236, 352, and 1488 are highlighted in wheat, whereas the β-ctCSP epitope of mAb 1512 is shown in pink. Residues that are present in both the α-ctCSP and β-ctCSP epitopes are highlighted in orange. The transparent surfaces overlaid with ribbons of ctCSP are also shown and the combined α-ctCSP epitopes, β-ctCSP epitope, and overlapping epitopes are colored in pink, wheat, and orange as above. (**B**) Dissociation constant ($K_D$) of all mAbs was measured against all generated ctCSP peptides using surface plasmon resonance (SPR). The strongest to weakest affinities are represented as a gradient from red to yellow and green, respectively. Boxes labeled with NB (no binding) represent haplotypes that had binding responses less than the negative control (humanized mAb 5D5, a *Pf*CSP N-terminus specific antibody).

The structures of α-ctCSP mAbs might offer hints on some of the binding activities observed in this class of antibodies. We first noted that, besides the wild-type 3D7 peptide, mAb234 binds to four additional peptides, H18, H1, H234, and H92 (Fig 3). The haplotype H234 differs substantially from 3D7 at the Th2R region with six mutations yet can still interact with mAb234 (Fig 3). The crystal structure suggests that the binding promiscuity of mAb234 to Th2R may be possible because the H-bonds between mAb234 and Th2R are mostly mediated by water molecules (S1 Fig). In fact, when we modeled replacement of the 3D7 residues in the crystal structure with H234 haplotype residues, we found that Fab234 can accommodate these mutations with no clashes and may be able to even make similar H-bonds with the mutated ctCSP (S6A Fig). The surface of mAb234 may also be large enough to accommodate some greater changes in a side chain, e.g., the N321K mutation (S6A Fig), although that needs to be experimentally confirmed. The limiting region seems to be Th3R even though it is difficult to describe why this is the case based only on our structure of Fab234 in complex with the 3D7 peptide. Based on the binding study, however, we observed that mAb234 cannot bind to haplotypes with D356N/E mutations and does not seem to bind to haplotypes with an E357 mutation that is not accompanied by mutation at A361 (S6B Fig). The combination of mutations in Th2R (or overall) may also play a role as the peptides H12 and H42 do not fall within the two criteria above, but still cannot interact with mAb234. Similar to mAb234, mAb236 is flexible for binding to Th2R as it also binds to the H234 haplotype (Fig 3B). However, binding to Th3R seems to be crucial to mAb236 since the antibody does not bind to any haplotypes that have mutations in that region (Fig 3A).

In contrast to the promiscuity in the mAb234 binding to the Th2R helix, the interactions with that region seem to be critical for mAbs 352 and 1488 as these two antibodies only bind the H18 haplotype (besides 3D7), which has no mutations in Th2R except for L327I in the perimeter (Fig 3). One explanation for this intolerance for Th2R mutations is that mAb352 and 1488 have tighter interacting surfaces that would clash with e.g., the N321K mutation, which is present in most haplotypes (S7 Fig). For haplotypes H36 and H92, which do not have N321K mutation (Fig 3A), we suspect that changes in residues 317 and 318 may disrupt the binding of ctCSP to mAb352 and 1488, although it is difficult to specify how this happens without determining structures of these antibody-peptide complexes (changes in residues 317 and 318 do not seem to obviously clash with the antibody surface).

## ctCSP specific mAb demonstrates anti-malarial inhibitory activity in a small animal model

To determine in vivo characteristics of β-ctCSP antibodies, we first performed immunofluorescence experiments with fixed 3D7 *P. falciparum* sporozoites (Fig 4). α-ctCSP and β-ctCSP antibodies 236 and 1512 showed similar staining patterns with fixed 3D7 *P. falciparum* sporozoites as mAb311, which is a previously characterized NANP repeat mAb [24] (Fig 4). These images are representative of a majority of all stained sporozoites. Of all organisms, approximately 30% remained unstained. Next, a parasite liver burden study in mice was conducted to determine in vivo functional activity conferred by representative α-ctCSP and β-ctCSP mAbs, 236 and 1512, respectively. Mice were passively transferred intravenously (IV) with the antibodies and challenged 16 h later with chimeric *P. berghei* sporozoites expressing full-length *P. falciparum* 3D7 CSP [41,42]. Mice were then injected with D-luciferin and imaged to measure bioluminescence in the infected livers. mAb 236 showed an average of 48% inhibition of parasite liver burden at 300 μg and 26% at 100 μg of antibodies when the mean percentage in the untreated control group was set as 0%. Statistical significance of mAb 236 at 300 μg and 100 μg was achieved compared to control ($p < 0.035$) using the Kruskal-Wallis test (Fig 5 and S8 Fig). mAb 1512 averaged 33%

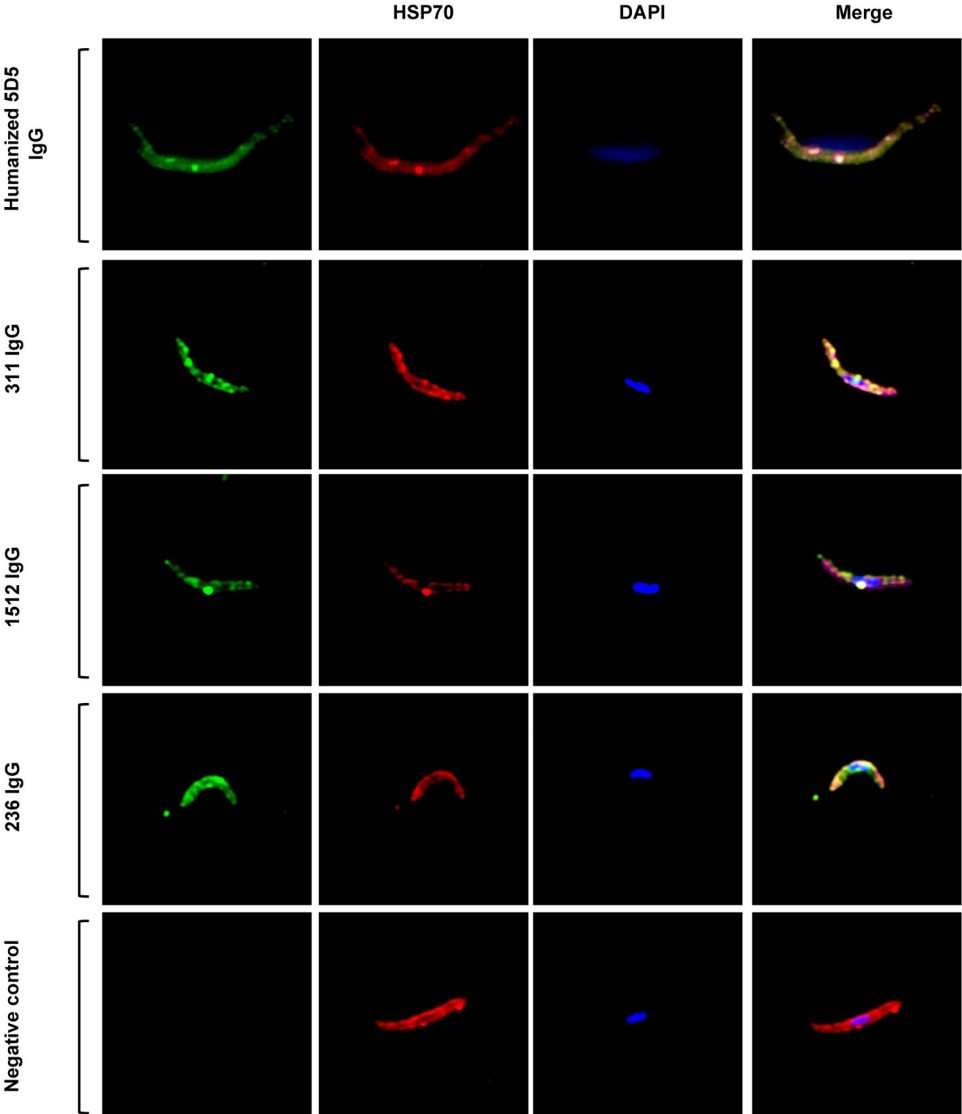

**Fig 4. Confocal microscopy images of *P. falciparum* sporozoites bound by anti-CSP antibodies.** *P. falciparum* sporozoites were dissected from infected mosquito salivary glands and purified. The binding of anti-*Pf*CSP mAbs specific to N-terminal region of *Pf*CSP (humanized mAb 5D5), NANP repeats (mAb311), β-ctCSP (mAb1512), and α-ctCSP (mAb236) was observed using the secondary antibody goat-anti-human Alexa 488 (green). A polyclonal anti-HSP70 antibody targeting *P. falciparum* heat shock protein 70 was used as a positive control and stained with the secondary antibody, goat anti-mouse Rhodamine Red-X (red). The sporozoite nuclei were stained with DAPI (blue). The merged images are also displayed. Images were acquired using a Zeiss LSM880 with Airyscan Confocal Microscope.

inhibition at 300 μg and 29% inhibition at 100 μg of antibodies, which both showed statistically significant protection ($p < 0.035$) compared to the control group (Figs 5 and S8). The range of % inhibition values in the mAb-treated animals was considerable. Of note, 4/15 1512-treated animals showed >75% inhibition of parasite liver burden.

## Discussion

In this study, we characterized anti-CSP C-terminus (ctCSP) antibodies derived from volunteers immunized with RTS,S/AS01 vaccine after either a standard or fractional third dose in a

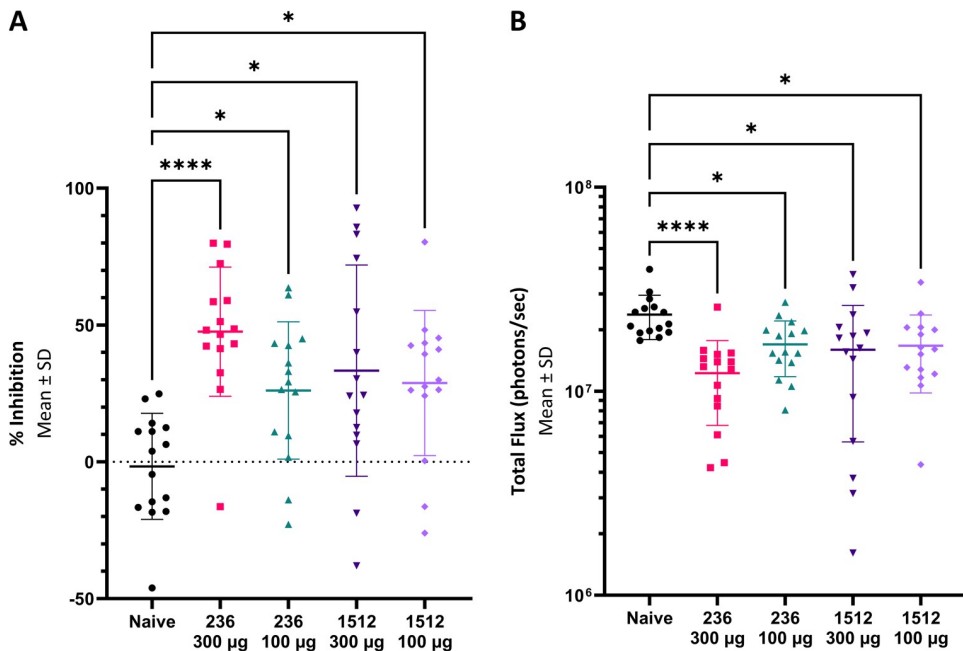

**Fig 5. Antibody inhibition of malaria infection in mice.** Mice, five per group in three separate experiments, were injected IV with the mAbs and 16h later challenged with chimeric *P. berghei* sporozoites expressing full-length *P. falciparum* CSP. Mice were injected with D-Luciferin and imaged with the IVIS spectrum to measure the bioluminescence expressed by the chimeric parasites. Data are presented as % inhibition of parasite burden in the liver (**A**) or as total flux (**B**) in each mouse as compared to the mean of the untreated control group (0%). In both data representations, mAb1512 at both 300 μg and 100 μg and mAb236 at both 300 μg and 100 μg exhibited significant inhibition compared to the control group (* = $p < 0.035$ or **** = $p < 0.0001$ by Kruskal-Wallis test).

phase 2a trial [36]. The possibility of two epitope regions in ctCSP was originally suggested by studies from Plassmeyer et al. [43]. Our competition assay with seven antibodies indeed indicated that two regions could be identified on ctCSP: one region recognized by mAbs 234, 236, 352, and 1488, and another targeted by mAbs 1512, 1504, and 1550. The non-reciprocal competition seen in these experiments suggests that binding of the 236-type mAbs to ctCSP inhibits binding of the 1512-type mAbs with the reverse not being the case, i.e. binding of 1512-type Abs does not affect 236-type Ab binding. Studies are underway to determine the basis of this non-reciprocal binding between these two groups of anti-ctCSP antibodies. Crystal structures showed these two epitope regions correspond to a previously identified region [37] containing the polymorphic Th3R and Th2R, which we termed here as the α-epitope region (α-ctCSP), and a novel β-epitope region (β-ctCSP) that consists of the CS.T3 and region II+ of *Pf*CSP, respectively. Of note, the region II+ has been shown to be conserved not only among *P. falciparum* strains but also across different *Plasmodium* species [44–46] (S9 Fig). We obtained crystal structures of five Fabs in complex with ctCSP, four of which bind to α-ctCSP. These four Fabs 234, 236, 352, and 1488 share a similar binding mode to α-ctCSP as the previously characterized mAb 1710, derived from a European donor immunized with live *Pf*SPZ under chloroquine prophylaxis [37]. This convergent binding mode involves CDRs L1 and L2 interaction with Th2R, CDRs H1 and H2 recognition of Th3R, and CDR H3 insertion into the conserved hydrophobic pocket in ctCSP (Fig 2). However, recognition of the polymorphic Th2R and Th3R by these mAbs is specific to the *P. falciparum* 3D7 strain upon which the RTS,S/AS01 vaccine is based. Thus, these mAbs exhibit poor or no binding to other naturally occurring strains, with limited binding restricted to the α-ctCSP sequences most similar to the 3D7 strain

(Figs 3 and S5). For example, mAb352 and 1488 bind only to haplotypes H18 and 3D7, which differ by only a single amino acid in the Th2R epitope. mAb236 binds only to haplotypes H234 and 3D7, which are identical in the Th3R epitope. Therefore, as previously suggested [37], diversity in the α-ctCSP epitope could significantly limit the breadth of antibody responses. In contrast, mAb1512 recognizes a novel epitope on ctCSP. This β-ctCSP epitope consists of the less variable CS.T3 and CSP region II+. The high conservation of the region II+ sequence is likely due to its key roles in sporozoite motility and infection [47,48]. Consistent with the conserved nature of this epitope, mAb1512 can bind with strong affinity to all 15 haplotypes of ctCSP that represent strains found across East and West Africa [31] (Figs 3 and S5). Thus, the β-ctCSP epitope could likely elicit broader antibody responses and represent a target for structure-guided vaccine design to protect against a wide range of *P. falciparum* strains and minimize antibody escape.

All anti-α-ctCSP antibodies show high affinity in the pM range against the matched 3D7 haplotype (Figs 3 and S5), and antibodies against β-ctCSP display high affinity in the nM to the pM range across all of the ctCSP haplotypes tested. In the liver burden assay in mice, a dose of 300 μg anti-α-ctCSP mAb236 exhibits functional activity corresponding to 48% inhibition of parasite burden after challenge with 3D7-expressing sporozoites, whereas the β-ctCSP mAb 1512 shows 33% inhibition at the same antibody dose (Fig 5). Of note, mAb 1512 appears to elicit similar in vivo functional activity as compared to mAb 236. However, the functional activity conferred by these ctCSP antibodies, either α- or β-epitope binders, is still considerably less than many potent anti-NANP-repeat antibodies [24,25,42] which usually display >90% inhibition of parasite burden at a 100 μg dose of antibody [25]. The functional activity results are intriguing as a recent report suggests that RTS,S protection in humans may be associated with the breadth of the ctCSP binding response [28]. However, the mouse infection model may not adequately represent the functional activity and rank order for antibodies directed to different regions of CSP in humans. It has been shown that the breadth of the ctCSP antibody response following RTS,S/AS01B immunization is strongly associated with protection as assessed by parasitemia [28]. Therefore, it is not unreasonable to hypothesize that the ctCSP mAbs here would also reduce parasitemia in the model.

Our confocal microscopy data show qualitatively that the α-ctCSP, β-ctCSP, and NANP repeat epitopes are accessible on the surface of salivary gland 3D7 *P. falciparum* sporozoites. Although these experiments utilized formaldehyde fixation, which under certain circumstances can alter the native conformation of surface proteins, it has been shown that the C-terminal domain on whole sporozoites is accessible to antibody binding in both fixed and unfixed states [37,43]. The moderate protection seen so far with ctCSP mAbs may be in part attributed to only one C-terminal antibody bound per *Pf*CSP compared to anti-NANP-repeat antibodies that are capable of binding to 10 or more sites within the NANP-repeat region of *Pf*CSP [33]. Our confocal microscopy data, complemented by our in-vivo findings, suggest these ctCSP epitopes are accessible on the surface of sporozoites, but whether ctCSP mAbs can achieve enhanced protection levels in combination with potent anti-NANP/junction mAbs in humans has yet to be explored.

The crystal structure also shows that 1512 utilizes a 23-amino-acid CDR H3 to interact with β-ctCSP. The low frequency of such long CDRH3s in the human antibody repertoire [49], may therefore present challenges in elicitation of the corresponding antibodies, especially to β-ctCSP. Interestingly, this structural feature is observed in many broadly neutralizing antibodies against the HIV-1 envelope glycoprotein, either to penetrate through the extensive glycan shield or target conserved epitopes that are located in less accessible regions, such as the membrane proximal external region (MPER) epitope close to the viral membrane [49]. Thus, future studies could focus on antibodies with a shorter CDRH3 [22] that bind to the β-ctCSP epitope,

such as mAb1550 reported here. As current vaccine designs and antibody characterization focus heavily on the NANP repeat region, exploring a diverse repertoire of non-repeat mAbs could reveal additional epitopes with potential additive or synergistic protective effects.

While prior studies involving volunteers vaccinated with live *Pf*SPZ under chloroquine prophylaxis suggested antibodies against ctCSP were rare in that setting, recent results have shown immunization with a CSP construct that has only 9 NANP repeats in mice induced lower overall B-cell responses to the NANP repeats, but comparatively stronger responses to non-repeat epitopes [50]. Furthermore, strong responses against NANP repeats also cause antibody feedback that limits the boosting of anti-repeat antibodies, but in turn, drives expansion of anti-ctCSP antibodies in subsequent boosts [51]. These results suggest the number of NANP repeats in next-generation vaccine constructs could be optimized to balance antibody responses against both NANP repeats and the C-terminal domain of *Pf*CSP. Finally, examination of correlates of protection against malaria following vaccination with RTS,S/AS01 reveals a statistically significant impact of C-terminal specific B cell and antibody responses that could be improved upon with increased affinity, breadth, and durability of the corresponding B cell responses [27,28].

Overall, our findings have revealed antibody binding and in vivo inhibitory activity to two ctCSP epitopes: a novel region on ctCSP (β-ctCSP epitope) that can elicit antibodies with markedly greater breadth against a large diversity of *P. falciparum* ctCSP variant peptides compared to antibodies to a previously defined ctCSP region (α-ctCSP epitope). These insights into the immunogenicity of ctCSP emphasizes that this domain could play an important role in examination of correlates of protection against malaria and help guide design of next-generation *Pf*CSP-based vaccines and medical countermeasures.

## Materials and methods

### Ethics statement

The assays using mice were performed in strict accordance with the recommendations in the Guide for the Care and Use of Laboratory Animals of the National Institutes of Health. The protocol was approved by the Animal Care and Use Committee of the Johns Hopkins University, protocol number MO18H419.

### Peptide and recombinant protein production

The biotinylated ctCSP Pf16 peptide (residues 283–375 of *P. falciparum* 3D7 CSP) was made by Biomatik (Wilmington, Delaware). The peptide was dissolved in DMSO to 20 mg/mL, then diluted in sterile ddH$_2$O to final assay concentration. Fifteen haplotypes of ctCSP peptides, harboring polymorphic amino acid positions exhibiting strong signals of balancing selection and/or observed to be associated with allele-specific vaccine protection in the Phase 3 ancillary genotyping study of the RTS,S/AS01 vaccine [31], were generated as described previously [28]. To evaluate the allelic breadth of antibody binding affinity, the haplotypes capture a variable degree of divergence from the 3D7 vaccine strain at the polymorphic Th2R and Th3R epitopes [28].

The αTSR domain or ctCSP for X-ray crystallography was constructed with residues 310–375 of *P. falciparum* 3D7 CSP, as defined in the previous study [15], followed by a 6xHis tag. The pET28a plasmid containing the ctCSP construct was transformed into *E. coli* SHUFFLE competent cells. A single colony was used to start a 50-mL overnight culture. 1-L culture was inoculated the next day with 25 mL overnight culture and was grown at 37˚C until the optical density at 600 nm reached ~0.6. The cultures were then induced with 0.5 mM isopropyl β-D-1-thiogalactopyranoside at 18˚C overnight. The cells were then harvested and lysed by

microfluidization (50 mL lysis buffer: 20 mM $Na_2HPO_4$ pH 7.2, 450 mM NaCl, 0.5% sarkosyl, and 1 Roche protease inhibitor tablet). The lysate was centrifuged, and the supernatant was supplemented to a final volume of 200 mL with 20 mM $Na_2HPO_4$ pH 7.2, and 450 mM NaCl and allowed to incubate overnight with 5-mL Ni cOmplete resin (Roche). After washing with 20 mM $Na_2HPO_4$ pH 7.2, 450 mM NaCl, 0.1% sarkosyl and another wash with 75 mM Tris pH 9.0, protein was eluted with 200 mM imidazole, 75 mM Tris pH 9.0, 150 mM NaCl. Fractions were concentrated and further purified by gel filtration on Superdex 200 16/90 (GE Healthcare).

## Antibody production

For crystallization, all Fabs were expressed in ExpiCHO cells and purified using a HiTrap Protein G HP column (GE Healthcare) followed by size exclusion chromatography (Superdex 200 16/90; GE Healthcare) in Tris Buffered Saline (TBS: 50 mM Tris pH 8.0, 137 mM NaCl, 3.6 mM KCl). For all in vivo and in vitro studies, mAbs were expressed in 293F cells and purified using Protein G Sepharose (GE Healthcare) in a gravity drip column.

## Epitope binning experiment

Epitope binning experiments were performed using bio-layer interferometry on Octet HTX (ForteBio, now Sartorius, Gottingen, Germany) at 25˚C. Biotinylated CSP was loaded onto streptavidin sensors (ForteBio) at 60 nM in Octet kinetic buffer (0.002% Tween 20 + 0.1 μg/ mL BSA in PBS). Loaded sensors were then dipped into the first saturating mAb (mAb1) at 20 μg/mL followed by the second competing mAb (mAb2) at 5 μg/mL. Competition experiments were performed with all seven antibodies, resulting in 49 pairs as in Fig 1B. BLI experiments were performed with the following steps: 1) baseline in kinetics buffer for 60 s; 2) loading of biotinylated *Pf*CSP for 120 s; 3) baseline for 60 s; 4) binding of saturating mAb1 for 600s; 5) baseline for 60 s; and 6) binding of competing mAb2 for 300 s. Octet binding responses were recorded as the change in nanometers (Δnm) of the shift in reflected light wavelengths from baseline due the accumulation of proteins at the sensor tip. Epitope competition was calculated for each mAb pair by calculating the percent inhibition of mAb2 binding in the presence of the saturating mAb1 as in a previous study [52]. The mAb2 control binding was calculated by averaging the observed Δnm of three non-competitive binding events of mAb2 at equilibrium. Percent inhibition was calculated as: 100 - [(Δnm from the mAb2 binding in competition/ Δnm from mAb2 maximum control binding) x 100].

## Crystallization, structure determination and analysis

Fabs 234, 236, 352,1488, and 1512 were concentrated to 10 mg/mL and mixed with ctCSP (residues 310–375 of PfCSP strain 3D7), in a 2:1 molar ratio of Fab to ctCSP. Each Fab-ctCSP complex was purified by size exclusion chromatography (Superdex 200 16/90; GE Healthcare). Crystal screening was carried out using our high-throughput, robotic CrystalMation system (Rigaku, Carlsbad, CA) at The Scripps Research Institute, which is based on the sitting drop vapor diffusion method, with 35 μL reservoir solution and each drop consisting of 0.1 μL protein + 0.1 μL precipitant. Fab234-ctCSP co-crystals were grown in 20% PEG-8000 and 0.1 M HEPES pH 7.5 at 20˚C and were cryoprotected in 20% ethylene glycol. Fab236-ctCSP crystals grew in 30% PEGME 2000, 0.2 M ammonium sulfate, and 0.1 M acetate pH 4.6 at 20˚C. Fab352-ctCSP crystals grew in 20% isopropanol, 20% PEG-4000, and 0.1 M sodium citrate pH 5.6 at 20˚C and were cryoprotected in 20% ethylene glycol. Fab1488-ctCSP crystals grew in 20% PEG 3350, and 0.2 M potassium chloride, pH 6.9 at 20˚C and were cryoprotected in 20% ethylene glycol. Fab1512-ctCSP crystals grew in 20% PEG-3000, 0.2 M sodium chloride, and

0.1 M HEPES pH 7.5 at 20°C and were cryoprotected in 20% ethylene glycol. X-ray diffraction data were collected at the Advanced Photon Source (APS) beamline 23ID-B, or at the Stanford Synchrotron Radiation Lightsource (SSRL) beamline 12–2, and processed and scaled using the HKL-2000 package [53]. The structures were determined by molecular replacement using Phaser [54]. Structure refinement was performed using phenix.refine [55] and iterations of refinement using Coot [56]. Amino-acid residues of the Fabs were numbered using the Kabat system, and the structures were validated using MolProbity [57]. For structural analysis, buried surface areas (BSAs) were calculated with the program MS [58], and hydrogen bonds were assessed with the program HBPLUS [59].

## Surface plasmon resonance

Affinity experiments were performed on a Biacore 8k at 25°C. All experiments were carried out with a flow rate of 30 μL/min in a mobile phase of HBS-EP+ [0.01 M HEPES (pH 7.4), 0.15 M NaCl, 3 mM EDTA, 0.0005% (v/v) Surfactant P20]. Anti-Human IgG (Fc) antibody (Cytiva 29234600) was immobilized via standard NHS/EDC coupling to a Series S CM-5 (Cytiva BR100530) sensor chip. Each mAb was injected over the chip followed by a wait period to normalize captured response units (RU). A concentration series of ctCSP peptides was injected across the antibody and control surface for 2 min, followed by 2000–10000 second dissociation phase. Regeneration of the surface in between injections of ctCSP peptides was achieved with a single, 120 s injection of 3 M $MgCl_2$. Kinetic analysis of each reference subtracted injection series was performed using the BIAEvaluation software (Cytiva). All sensorgram series were fit to a 1:1 (Langmuir) binding model of interaction. Sensorgrams were graphed using R (v4.05)

## Whole sporozoite immunofluorescence and confocal microscopy

*P. falciparum*-infected *A. stephensi* mosquitos were obtained from Johns Hopkins Bloomberg School of Public Health Parasitology Core Facility. *P. falciparum* sporozoites were obtained by dissection of infected *Anopheles stephensi* mosquito salivary glands. Dissected salivary glands were homogenized in a glass tissue grinder and filtered twice through nylon cell strainers (Millipore SCNY00020) and counted using a Neubauer hemocytometer. The sporozoites were kept on ice until needed. Purified *P. falciparum* sporozoites were fixed with PFA 4%, permeabilized with 0.1%-Triton X-100/PBS 0.1%, blocked with 1% BSA-PBS 1x, and incubated with the primary human monoclonal antibodies: mAb5D5, mAb311, mAb1512, and mAb236 at 4°C; as a positive control, mouse anti-HSP70 polyclonal antibody targeting *P. falciparum* heat shock protein was used (GeneScript, ID U4463DB220-1), and pre-immune sera was used as a negative control (GeneScript, ID U4463DB220-2), dilution 1:500 (stock 1 μg/ul). Parasites were incubated with the secondary antibodies: Alexa Fluor 488 AffiniPure Goat Anti-Human IgG, Fcγ fragment specific (Jackson ImmunoResearch Labs, Cat No. 109545098), or Rhodamine Red-X (RRX) AffiniPure Goat Anti-Mouse IgG, Fcγ fragment specific (Jackson ImmunoResearch Labs, Cat No. 115295071), dilution 1:1,000. The nucleus was stained with Vectashield with DAPI (Vector Lab, H-1500). The images were acquired using a Zeiss LSM880 with Airyscan Confocal Microscope (63x oil immersion lens); diode laser power was set to 3% for 405, 488, 561, and 640 nm. The images were captured and processed using the confocal ZEN software (Black edition Zeiss).

## Parasite liver burden assay to assess antibody protection against malaria

Parasite liver burden assays were performed as described previously [41,42]. Briefly, female, 6–8 weeks old C57Bl/6 mice were purchased from Charles River. To measure liver burden, mice ($N = 5$) were IV injected with 100 or 300 μg of Ab per mouse as indicated and, 16 h later,

challenged IV with 2000 *P. berghei* transgenic sporozoites expressing *P. falciparum* CSP and luciferase. 42 h after challenge, mice were injected IP with 100 μL of D-luciferin (30 mg/mL), having been anesthetized by exposure to isoflurane. Bioluminescence in the liver was measured using an IVIS Spectrum (Perkin Elmer, Waltham, MA).

### Statistical analysis

The parasite liver burden load data ($N = 5$ mice) in three separate experiments were compared for significance using the Kruskal-Wallis test, with a single pooled variance, where $p < 0.035$ (*) or $p < 0.0001$ (****) indicated levels of statistically significant differences. Both the % inhibition and the total flux data were reported as the mean ± SD.

### Supporting information

**S1 Fig. Hydrogen bonding networks between mAbs 234, 236, 352, 1488 and ctCSP.** Hydrogen bonds between mAbs 234, 236, 352, 1488 and ctCSP are shown. Antibodies and ctCSP are shown in a ribbon representation, with side chains as sticks. ctCSP is colored green, salmon, and magenta for the alpha helix, CS flap, and linker region, respectively (see also Fig 2B). Antibody heavy and light chains are colored orange and yellow, respectively. Black dashes represent hydrogen bonds, whereas oxygen atoms that represent water molecules are shown as red spheres.
(TIF)

**S2 Fig. Individual residue contributions to the Buried Surface Area (BSA) at the Fab-peptide interface.** BSAs are shown in yellow bars for the heavy and light chains of (**A**) Fab234, (**B**) Fab236, and (**C**) Fab352. CDRs are colored in green, blue, magenta for CDR H1, H2, H3 for heavy chains or L1, L2, and L3 for light chains, respectively. Additionally, the alignment between the Fab heavy and light chain sequences and germline *IGHV* and *IGLV* gene sequences, respectively, indicates which residues are somatically mutated.
(TIF)

**S3 Fig. Individual residue contributions to the Buried Surface Area (BSA) at the Fab-peptide interface.** BSAs are shown in yellow bars for the heavy and light chains of (**A**) Fab1488 and (**B**) Fab1512. CDRs are colored in green, blue, magenta for CDR H1, H2, H3 for heavy chains or L1, L2, and L3 for light chains, respectively. Additionally, the alignment between the Fab heavy and light chain sequences and germline *IGHV* and *IGLV* or *IGKV* gene sequences, respectively, indicates which residues are somatically mutated.
(TIF)

**S4 Fig. Hydrogen bonding networks between mAb 1512 and ctCSP.** Hydrogen bonds between mAb1512 and ctCSP are shown. The antibody and ctCSP are shown in ribbon representation, with side chains as sticks. ctCSP is colored green and magenta for the alpha helix and CS flap regions, and different shades of blue for the three TSR homology region strands (see also Fig 2B). Antibody heavy and light chains are colored orange and yellow, respectively. Black dashes represent hydrogen bonds, whereas oxygen atoms that represent water molecules are shown as red spheres.
(TIF)

**S5 Fig. Binding of anti-CSP antibodies to ctCSP haplotypes measured by surface plasmon resonance (SPR).** Panel of the binding of ctCSP peptides to mAbs via a Fc-capture, single cycle, multi-injection method. Association and dissociation constants were calculated through a 1:1 Langmuir binding model using BIAevaluation software. NB = No binding. SPR

sensorgrams for ctCSP peptide binding displaying best global fits. Antibodies were captured on anti-human IgG (Fc) antibody immobilized on a CM5 chip and varying concentrations of ctCSP were injected using a single cycle method. Sensorgrams in resonance units (RUs) plotted against time are shown.
(TIF)

**S6 Fig. Analysis of the binding of mAb234 to different haplotypes.** (**A**) The side chains of residues in the crystal structure of Fab234 in complex with the 3D7 peptide were manually mutated to the side chains of H234 in PyMOL. Top panel: the crystal structure of Fab234 in complex with 3D7 ctCSP (colored and represented as in S1 Fig) is overlaid with the computationally mutated H234 ctCSP (grey ribbons with side chains as sticks). The residues are labeled in the format of A/B where A is an H234 residue and B is a 3D7 residue. Bottom panel: Fab234 (white surfaces) in complex with the computationally mutated H234 ctCSP (colored and represented as in S1 Fig). (**B**) Table showing successive filtering of all peptides based on two criteria. The first column lists all peptides. The second column lists all peptides that has no mutation in residue D356, and the last column lists all peptides that has E357Q mutation with a mutation in residue A361. All peptides in last column, except for, H12 and H92 (in red), binds to mAb234 (see also Fig 3).
(TIF)

**S7 Fig. Analysis of the binding of mAb352 and 1488 to the N321K mutation.** The side chain of N321 in the crystal structure each Fab in complex with the 3D7 peptide was manually mutated in PyMOL to K321, which is found in some haplotypes, The antibody is shown as white surfaces, whereas the Th2R alpha-helix is shown in ribbons with side chains as sticks.
(TIF)

**S8 Fig. Statistical analysis of antibody inhibition of malaria infection in mice (see also Fig 5).** Mice, five per group in three separate experiments, were examined for % inhibition, which is the percentage of parasite burden in the liver in each mouse as compared to the mean of the untreated control group (100%). The bars in Fig 5 represent geometric mean. mAb1512 at both 300 μg and 100 μg and mAb236 at 300 μg and 100 μg exhibited statistically significant inhibition compared to the control group ($p < 0.035$ by Kruskal-Wallis test). Mean rank difference represents the difference in the geometric mean between naive and mAb.
(TIF)

**S9 Fig. Diversity of the C-terminal domain of *Pf*CSP from different species.** The alignment of ctCSP *Pf*CSP sequences from different species was adapted from Doud *et al.* [15]. The Th2R, RII+, Th3R, and CS.T3 regions are enclosed in green, cyan, magenta, and blue boxes respectively. The combined α-ctCSP epitopes from mAbs 234, 236, 352, and 1488 are highlighted in yellow, whereas the β-ctCSP epitope of mAb 1512 is shown in pink. Human *Plasmodium* parasites: *P. falciparum*, *P. vivax*, *P. malariae*, and *P. knowlesi* (also infects non-human primates); non-human primate *Plasmodium* parasites: *P. reichenowi*, *P. simium*, and *P. cynomolgi*; avian *Plasmodium* parasite: *P. gallinaceum*; mouse *Plasmodium* parasites: *P. yoelii* and *P. berghei*.
(TIF)

**S1 Table. X-ray data collection and refinement statistics.**
(DOCX)

**S2 Table. Buried surface area (BSA) and CDRH3 length of mAbs analyzed in this study.**
(DOCX)

## Acknowledgments

We are grateful to the staff of Stanford Synchrotron Radiation Lightsource (SSRL) Beamline 12–2 and Advanced Proton Source (APS) beamline 23ID-B for assistance. Use of the SSRL, SLAC National Accelerator Laboratory, is supported by the U.S. Department of Energy, Office of Science, Office of Basic Energy Sciences under Contract No. DE-AC02–76SF00515. The SSRL Structural Molecular Biology Program is supported by the DOE Office of Biological and Environmental Research, and by the National Institutes of Health, National Institute of General Medical Sciences (including P41GM103393). This research used resources of the Advanced Photon Source, a U.S. Department of Energy (DOE) Office of Science User Facility operated for the DOE Office of Science by Argonne National Laboratory under Contract No. DE-AC02-06CH11357. Material has been reviewed by WRAIR and USAID and there is no objection to its presentation and/or publication. The opinions or assertions contained herein are the private views of the authors, and are not to be construed as official, or as reflecting views of the Department of the Army, the Department of Defense, or USAID.

## Author Contributions

**Conceptualization:** Nathan Beutler, Tossapol Pholcharee, David Oyen, C. Richter King, Fidel Zavala, Dennis R. Burton, Ian A. Wilson, Thomas F. Rogers.

**Formal analysis:** Nathan Beutler, Tossapol Pholcharee, Yevel Flores-Garcia, Randall S. MacGill, Elizabeth A. Winzeler, C. Richter King, Fidel Zavala, Dennis R. Burton, Ian A. Wilson, Thomas F. Rogers.

**Investigation:** Nathan Beutler, David Oyen, Yevel Flores-Garcia, Randall S. MacGill, Elijah Garcia, Jaeson Calla, Mara Parren, Linlin Yang, Emily Locke, Jason A. Regules, Sheetij Dutta, Fidel Zavala.

**Resources:** Wayne Volkmuth, Emily Locke, Daniel Emerling, Angela M. Early, Daniel E. Neafsey, C. Richter King.

**Writing – original draft:** Nathan Beutler, Tossapol Pholcharee, Dennis R. Burton, Ian A. Wilson, Thomas F. Rogers.

**Writing – review & editing:** Nathan Beutler, Tossapol Pholcharee, David Oyen, Yevel Flores-Garcia, Randall S. MacGill, Elijah Garcia, Jaeson Calla, Mara Parren, Linlin Yang, Wayne Volkmuth, Emily Locke, Jason A. Regules, Sheetij Dutta, Daniel Emerling, Angela M. Early, Daniel E. Neafsey, Elizabeth A. Winzeler, C. Richter King, Fidel Zavala, Dennis R. Burton, Ian A. Wilson, Thomas F. Rogers.

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
