## [Decision Letter · Decision Letter 0]

22 Nov 2021

Dear Prof. Wilson,

Thank you very much for submitting your manuscript "A novel CSP C-terminal epitope targeted by an antibody with protective activity against Plasmodium falciparum." for consideration at PLOS Pathogens. As with all papers reviewed by the journal, your manuscript was reviewed by members of the editorial board and by several independent reviewers. The reviewers appreciated the attention to an important topic. Based on the reviews, we are likely to accept this manuscript for publication, providing that you modify the manuscript according to the review recommendations.

I am returning your manuscript with three reviews. The reviewers came to different conclusions about the paper, as you will see. After reading the reviews and looking at the manuscript, I have decided that the further mouse experiments requested by reviewer 2 are not necessary for this manuscript to meet the criteria for publication at PLOS Pathogens. There are, however, a few remaining minor revisions that need to be addressed to prepare the manuscript for publication.

Please pay particular attention to the following reviewer suggestions and give them due consideration.

• Please describe the magnitude of the polyclonal response to ctCSP compared to NANP repeat in the samples studied and relative frequency of anti ctCSP mAbs compared to anti NANP mAbs.

• The authors speculate that conformational changes induced by α-ctCSP mAb binding may hinder β-ctCSP mAb recognition. This does not appear to be currently supported by the structural studies in the manuscript. Please provide either more discussion highlighting that the claim is not based on structural data or additional structural information to support this claim?

• Is blood stage parasitemia prevented in any of the mice treated with mAbs 236 and 1512? If unlikely, can the authors please make this clear in the discussion in comparisons to expectations from previous results.

Sincerely,

Waihong Tham, Ph.D

Guest Editor

PLOS Pathogens

Eleanor Riley

Section Editor

PLOS Pathogens

Kasturi Haldar

Editor-in-Chief

PLOS Pathogens

orcid.org/0000-0001-5065-158X

Michael Malim

Editor-in-Chief

PLOS Pathogens

orcid.org/0000-0002-7699-2064

I am returning your manuscript with three reviews. The reviewers came to different conclusions about the paper, as you will see. After reading the reviews and looking at the manuscript, I have decided that the further mouse experiments requested by reviewer 2 are not necessary for this manuscript to meet the criteria for publication at PLOS Pathogens. There are, however, a few remaining minor revisions that need to be addressed to prepare the manuscript for publication.

Please pay particular attention to the following reviewer suggestions and give them due consideration.

• Please describe the magnitude of the polyclonal response to ctCSP compared to NANP repeat in the samples studied and relative frequency of anti ctCSP mAbs compared to anti NANP mAbs.

• The authors speculate that conformational changes induced by α-ctCSP mAb binding may hinder β-ctCSP mAb recognition. This does not appear to be currently supported by the structural studies in the manuscript. Please provide either more discussion highlighting that the claim is not based on structural data or additional structural information to support this claim?

• Is blood stage parasitemia prevented in any of the mice treated with mAbs 236 and 1512? If unlikely, can the authors please make this clear in the discussion in comparisons to expectations from previous results.

Reviewer Comments (if any, and for reference):

Reviewer's Responses to Questions

**Part I - Summary**

Reviewer #1: In this study, Beutler et al. characterize seven monoclonal antibodies (mAbs) specific for the P. falciparum C-terminal domain of CSP (ctCSP) that were isolated from volunteers immunized with RTS,S/AS01. Through biolayer interferometry competition binding assays and X-ray crystallography, the authors describe two distinct epitopes in the ctCSP αTSR domain: a previously identified epitope, denoted here as the α-ctCSP epitope, consisting of the polymorphic Th2R and Th3R regions, and a novel “β-ctCSP” epitope on the opposite face, which comprises conserved RII+ and CS.T3 sites. Next, using surface plasmon resonance, the authors examine the breadth of antibody binding to a panel of 15 ctCSP peptides generated to reflect the polymorphic variability of different P. falciparum isolates. Their results show that while binding of α-ctCSP-specific mAbs was limited to only two or five peptides, all three β-ctCSP-targeting mAbs demonstrated high affinity binding to every peptide in the panel. Finally, the authors evaluate the functionality of two representative ctCSP mAbs, revealing that these mAbs are able to access and bind their epitopes on the sporozoite surface via confocal microscopy, and exhibit limited inhibition of liver burden in a transgenic parasite mouse model.

Overall, this investigation identifies a conserved, unprecedented ctCSP epitope through extensive structural and biophysical characterization. In comparison to the central repeat region, the antibody response against ctCSP is much less well-studied. Indeed, only one other ctCSP-specific mAb had been structurally and functionally characterized to date. Thus, the novelty of this study and the importance of developing a more comprehensive understanding of the antibody response against all CSP domains warrants publication of this study, provided that the authors address the following concerns:

Reviewer #2: This paper describes the antigenicity, structures and anti-malarial inhibitory activity of antibodies derived from patients immunized with the RTS,S/AS01 vaccine. The authors determine 2 major distinct epitopes on the C-terminus of PfCSP and identify a novel epitope called the beta epitope region, which is more conserved than the alpha epitope region. The mAbs that target the beta epitope show cross-binding reactivity. Both the alpa and beta-epitope directed Mab show some anti-malarial inhibitory activity in mice.

This study is clear and concise and the paper well written. It should be of interest for the malaria vaccine field as there have been some controversies regarding the role of PfCSP C-terminus in the immune response.

Reviewer #3: The manuscript by Beutler et al., characterizes seven human mAbs specific to the CSP C-terminal domain derived from volunteers immunized with RTS,S/AS01. Competition binning experiments revealed that they bound to two distinct sites on ctCSP. The authors then determined the crystal structures of five of the Fab-CtCSP co-complexes, which revealed that the antibodies bind on opposite faces of ctCSP, a polymorphic epitope previously described and a novel non-polymorphic epitope. SPR against a diverse panel of CtCSP peptides confirmed that mAbs that recognized the non-polymorphic epitope were broadly reactive compared to the other mAbs. Finally, the authors tested the ability of two mAbs, one specific for each site described, to inhibit malaria infection in a murine model and showed 48% and 33% functional inhibition at the highest concentration tested.

Overall, this is a well designed and important study for the field. The crystal structures are well described and of excellent quality. My main concern surrounds the anti-malarial inhibitory data. As the authors concede, the two antibodies tested in the mouse model at 300ug, 236 and 1512, only inhibited functional activity by 48% and 33%, respectively. Based on previous studies from the same group (Flores-Garcia Y et al., 2019; Raghunandan R et al., 2020), it appears unlikely that blood stage parasitemia would be prevented in any of the mice, although this is not shown. This should be made clear in the discussion and is consistent with previous results. This is despite the Abs having picomolar affinity. An argument could be made that this study supports the notion that the ctCSP domain should de-prioritized in vaccine design, due to poor efficacy. However, I think the authors are fair in their discussion of the results and as they suggest, further experiments (beyond the scope of this MS) are needed in different model systems to show this definitively.

**Part II – Major Issues: Key Experiments Required for Acceptance**

Reviewer #1: Major points:

1) It would be important to state the way the ctCPS mAbs were isolated (from MBCs or PBs or PCs), the number of mAbs that were screened, and the down-selection process, if any. If possible, it would also be informative to understand the magnitude of the polyclonal response to ctCSP compared to the NANP repeat in the samples studied.

2) The confocal microscopy section of the study should be improved. For example, Fig 4 only shows staining data for a single parasite with each mAb. The data as presented does not allow to determine whether this staining pattern is the only one observed, or whether there was heterogeneity in the reactivity. If indeed heterogeneity was observed, it would be valuable to know whether the pattern described represents the majority or minority of PfCSP reactogenicity.

3) In addition, based on the methods provided, it appears that the sporozoites characterized are fixed and permeabilized - 4% PFA is standard fixative. Strong DAPI staining in Fig 4 is another strong indication that the cell membrane is no longer intact and that the sporozoites imaged are likely dead. As such, the mention of “live sporozoites” in lines 258-260 is inadequate, and should be described as “fixed sporozoites”. If indeed the sporozoites analyzed and imaged are believed to be “live”, it will be important to provide additional data to support such a claim.

4) Through competition binding studies, the α-ctCSP-targeting mAbs presented in this study were found to inhibit β-ctCSP mAb binding, but the same cannot be said for the reverse scenario. The authors speculate that this may be due to conformational changes induced by α-ctCSP mAb binding that hinder β-ctCSP mAb recognition (lines 176-179). Can the authors comment on whether their structural studies support this hypothesis? Do the crystallized αTSR domains overlay similarly regardless of the mAb bound? Conversely, could there be a different explanation for this data? e.g. partial occlusion of epitope because of preferred orientation of antigen on biosensors, or difference in on-rate/off-rate between mAbs 1 and 2?

Reviewer #2: The study would benefit from additional clarification and/or experiments:

- previous study by Scally at al indicated that Mabs directed against the alpha epitope region were not protective while this study show some level of inhibition with mAb236. It would be nice to repeat the in vivo protection with Mab 1710 as control to be able to compare side by side the reported findings. Additionally, adding a control anti-NANP mAb such as 317 will be of interest to compare side by side the inhibitory activity of mAbs directed against the repeat vs the C-terminus of PfCSP, even though the authors mention in the discussion that the inhibitory activity of the C-term MAb is less than that of the Mab directed against the repeat.

- Can the authors use the structures of the mAb to the alpha epitope region to explain differences observed in binding recognition to the different peptides

- Have the authors tried to co-crystallize 1504 and 1550? Can they mention why these structures are not shown here (there are 4 for one group vs 1 for the others), especially since one 1550 has the shorter CDRH3

Reviewer #3: (No Response)

**Part III – Minor Issues: Editorial and Data Presentation Modifications**

Reviewer #1: Minor points:

1) Impressively, the authors describe the crystal structures of five Fabs each in complex with the ctCSP αTSR. Does the breadth of binding of these mAbs against the panel of peptides tested by SPR correspond to the specific molecular interactions identified in the crystal structures? Providing a brief structural basis for the binding patterns across sequence diversity would further strengthen and solidify the message of this study.

2) In Fig 3a, a description of the orange-highlighted residues in the figure caption would help clarify the figure further.

3) Lines 111-113: this sentence suggests that 95% protective efficacy was observed in the control group of the Phase 2 clinical study examining vaccine candidate R21.

4) Lines 131 and 138: once defined, the authors should use acronyms consistently throughout the publication (i.e. “ctCSP” and “mAbs”)

5) Line 149-150: given that the study characterizes only one β-ctCSP-specific mAb with limited protective efficacy, I suggest that the authors soften the language of this statement to avoid over-generalizing their findings.

6) Line 158: it is unclear what peptide “NANPx” is referring to.

7) Line 214: it would be beneficial to comment on the conservation of the CS.T3 epitope in P. falciparum isolates not included in the study as well.

8) Line 240: it is unclear why reference 36 is called out in this sentence.

9) Lines 315 and 341: the authors are encouraged to streamline the Discussion section to avoid making the same statements twice. In this aspect, it would be helpful to the reader to better understand the inhibition range of the assay by mAbs if the liver burden activity of such an anti-NANP-repeat antibody could be shown directly in Fig. 5 along with the ctCSP antibodies.

10) Lines 326-327: The claim about the low frequency of long CDRH3s needs to be supported by a reference.

11) Line 373: Have the authors looked at whether all peptides are equally well folded (e.g. by circular dichroism) and therefore would be adequately recognized by antibodies that have conformational epitopes?

12) Line 413: nm should be used here rather than nM.

13) Line 430: “Advanced Photon Source”

14) Line 755: “Immunogenetics of ctCSP-specific mAbs”

15) Fig 1a: The first light chain V gene of mAb 352 is missing the “I” in “IGLV”. Also, what is the difference between groups labeled Fx017M and Fx017?

16) Fig 5: Assuming that the lines for each group indicate the mean, because the data are presented as % inhibition of parasite burden compared to the mean of the control group, shouldn’t the line for the control group align with the dotted line at 0%?

17) Figs S2 and S3 are missing y-axis titles, and it is also unclear why these plots have been grouped into separate figures.

18) Fig S4 caption: “Hydrogen bonding networks between mAb 1512 and ctCSP”

Reviewer #2: The authors mention that antibodies in mab 236 group could induce conformational changes, preventing binding of the beta epitope regions directed mabs but the structures don't see to show this. Can the authors add some discussion of what they think may be happening (line 176—179).

The authors should refer to PfCP and not CSP.

In Fig 2A, it will be nice to color code ctCSP as in figure 2B for one of the structure.

In figure 3, it will be nice to have a surface representation showing the overlapping region and domain described in 3A (maybe 2 views with 180 deg rotation).

Fig S2 - lacking Y axis legend

Reviewer #3: Can the authors provide the relative frequency of anti ctCSP mAbs compared to anti NANP mAbs in volunteers immunized with RTS,S/AS01? Are ctCSP rare or abundant?

Line 176. The authors suggest that upon 236, 234, 352, or 1488 mAb binding to ctCSP, conformational changes may be induced to prevent binding of 1512 etc. This is potentially very interesting. Do the crystal structures support this?

Line 209 – The authors state that mAb1488 has a lower affinity for ctCSP compared to the other mAbs, but this hasn’t been shown yet. Perhaps reference figure 3?

Line 248 – Based on the crystal structures, can the authors infer why mAb 234 showed greater breadth than the other a-ctCSP specific mAbs?

Line 386 – Incubate not incubated

Line 388 – protein was eluted “with” 200 mM

In Fig 2C, some of the aa’s are mis-labeled. Please check.

Fab234 FH101 – FH100A and HH102 – HH100B and SL32 not labeled

Fab236 PH100 not labeled

Fab 1488 – Label residue H56

Fab 1710 – Label SH54? Label LCDR3 residues

Figure 5 - It would be beneficial to see the data represented as total flux (photons/s) as well as inhibition, similar to Oyen D et al., 2020 Plos Path.

S1 Table – need to superscript some of the a

In Fab236-ctCSP column, Unique reflections (1,678)

In Fab1512-ctCSP column, No. of atoms ctCSP 1,004

PLOS authors have the option to publish the peer review history of their article (what does this mean?). If published, this will include your full peer review and any attached files.

Reviewer #1: No

Reviewer #2: No

Reviewer #3: No

Figure Files:

Data Requirements:

Reproducibility:

References:

---

## [Decision Letter · Decision Letter 1]

10 Feb 2022

Dear Prof. Wilson,

Thank you very much for submitting your manuscript "A novel CSP C-terminal epitope targeted by an antibody with protective activity against Plasmodium falciparum." for consideration at PLOS Pathogens. As with all papers reviewed by the journal, your manuscript was reviewed by members of the editorial board and by several independent reviewers. The reviewers appreciated the attention to an important topic. Based on the reviews, we are likely to accept this manuscript for publication, providing that you modify the manuscript according to the review recommendations. We hope that you will be able to return these changes by 14 days.

We would like to see textual changes and addition of the raw data for total flux as requested by Reviewer 1. In particular, as there are no additional experiments required we do ask that the authors pay particular attention to the textual changes that has been requested by Reviewer 1. Below we provide a quick summary of the changes that should be addressed.

1. The authors should caveat in the Discussion that their sporozoite binding studies are performed with fixed sporozoites, and speculate on how epitope accessibility/reactivity might differ in live sporozoites, which ultimately is more relevant.

2. If you are unable to show that the crystallised TSR domains are conformationally different upon mAb binding, please remove the hypothesis around conformational change in ctCSP induced by α-ctCSP-targeting mAbs as requested.

3. Please provide the raw data for the total flux (photons) as you have previously done for Oyen et al 2020.

Please prepare and submit your revised manuscript within 14 days. If you anticipate any delay, please let us know the expected resubmission date by replying to this email.

Sincerely,

Waihong Tham, Ph.D

Guest Editor

PLOS Pathogens

Eleanor Riley

Section Editor

PLOS Pathogens

Kasturi Haldar

Editor-in-Chief

PLOS Pathogens

orcid.org/0000-0001-5065-158X

Michael Malim

Editor-in-Chief

PLOS Pathogens

orcid.org/0000-0002-7699-2064

We would like to see aadditional textual changes and addition of the raw data for total flux as requested by Reviewer 1. In particular, as there are no additional experiments required we do ask that the authors pay particular attention to the textual changes that has been requested by Reviewer 1. We hope that you will be able to return these changes by two weeks.

1. The authors should caveat in the Discussion that their sporozoite binding studies are performed with fixed sporozoites, and speculate on how epitope accessibility/reactivity might differ in live sporozoites, which ultimately is more relevant.

2. If you are unable to show that the crystallised TSR domains are conformationally different upon mAb binding, please remove the hypothesis around conformational change in ctCSP induced by α-ctCSP-targeting mAbs as requested.

3. Please provide the raw data for the total flux (photons) as you have previously done for Oyen et al 2020.

Reviewer Comments (if any, and for reference):

Reviewer's Responses to Questions

**Part I - Summary**

Reviewer #1: The authors have revised their manuscript in accordance with the Reviewers’ comments.

Reviewer #3: The authors have addressed my concerns adequately.

**Part II – Major Issues: Key Experiments Required for Acceptance**

Reviewer #1: However, new insights in their responses have raised a need to further address major concerns to improve the quality of this manuscript:

1. The authors now mention a relatively high degree of heterogeneity observed amongst imaged sporozoites for reactivity by ctCSP-targeting mAbs. Additional figure panels quantifying the signal intensity across the sporozoite population and the percentage of stained sporozoites in each sample group would further strengthen their claims of epitope accessibility. In addition, the authors should caveat in the Discussion that their sporozoite binding studies are performed with fixed sporozoites, and speculate on how epitope accessibility/reactivity might differ in live sporozoites, which ultimately is more relevant. This concept is particularly critical given that a previous report had indicated much lower reactivity for a ctCSP-targeting mAb in the context of live sporozoites (Scally et al).

2. The authors have moved their speculations regarding the possibility of a conformational change in ctCSP induced by α-ctCSP-targeting mAbs to the Discussion section and acknowledged that additional structural studies to validate this hypothesis will be needed and have yet to be performed. Nonetheless, are there any insights in this regard already gained from the structural data presented in the current investigation i.e. do the crystallized αTSR domains overlay similarly regardless of the mAb bound? If this is the case, and limited evidence currently exists for this hypothesis, it would be appropriate to additionally present the hypothesis that the observed non-reciprocal phenomenon may just as well be due to the methodological and experimental set up. Or simply remove the hypothesis around conformational change in ctCSP induced by α-ctCSP-targeting mAbs if all crystallized αTSR domains overlay similarly regardless of the mAb bound.

3. In response to Reviewer 3’s request to add the raw data represented as total flux (photons/s) as they had previously done in Oyen D et al., 2020 Plos Path, the authors respond: “We feel that showing the total flux in log scale will not emphasize the difference from control and would prefer to keep the current representation in Fig. 5.” Although they may choose how best to represent the data in the main Figure, it is this Reviewer’s opinion that the authors should minimally present the raw data represented as total flux (photons/s) as they had done in a previous peer-reviewed publication as a Supplementary Figure as requested, to make the raw data available to the reader for critical analysis of the findings reported.

Reviewer #3: (No Response)

**Part III – Minor Issues: Editorial and Data Presentation Modifications**

Reviewer #1: Minor points:

1. Line 111-112: “Efficacy levels of 74% and 77% in the low- and high-dose adjuvant groups compared to the control group, respectively, were observed…”

2. Line 114: I believe reference #11 should be cited at the end of this sentence, rather than #12.

3. Line 275: “The surface of mAb234 may also be large enough”

4. Line 284-285: based on the crystal structure, are there any insights into the intolerance of mAb 236 for variation in the Th3R site?

5. Line 377-378: the possibility of synergistic effects of ctCSP mAbs combined with anti-NANP/junction mAbs was discussed in a recent publication by Wang et al. in Plos Pathogens (Dec 2021). The authors may want to comment on their findings in the context of this study to stay current with the field.

6. Fig 1A caption: “Immunogenetics of ctCSP-specific mAbs”

7. Fig S6A: labels of residue E314 referring to the H234 haplotype should be corrected to Q314. Numbering the sequence alignment in Fig 3A according to the full CSP protein sequence to match the labelling in Figs S6 and S7 would also be helpful to the reader.

8. Fig S7 caption: “The side chain of N321 in the crystal structure of each Fab…” – this sentence should also end with a period rather than a comma.

9. Fig S8: a description of the dark orange-highlighted residues in this figure caption would help to clarify the figure further.

Reviewer #3: I find the new discussion between lines 368 and 375 a little confusing. There are two sentences that both say that the breadth of the ctCSP response is associated with protection following RTS,S immunization. Perhaps if you remove one of them it would read better.

PLOS authors have the option to publish the peer review history of their article (what does this mean?). If published, this will include your full peer review and any attached files.

Reviewer #1: No

Reviewer #3: No

Figure Files:

Data Requirements:

Reproducibility:

References:

---

## [Editor Report · Decision Letter 2]

2 Mar 2022

Dear Prof. Wilson,

I apologize for the short delay in progressing your manuscript,. This was due to unforeseen circumstances affecting a member of the editorial team.

We are pleased to inform you that your manuscript 'A novel CSP C-terminal epitope targeted by an antibody with protective activity against Plasmodium falciparum.' has been provisionally accepted for publication in PLOS Pathogens.

Best regards,

Eleanor M. Riley

Section Editor

PLOS Pathogens

Eleanor Riley

Section Editor

PLOS Pathogens

Kasturi Haldar

Editor-in-Chief

PLOS Pathogens

orcid.org/0000-0001-5065-158X

Michael Malim

Editor-in-Chief

PLOS Pathogens

orcid.org/0000-0002-7699-2064
---

## [Editor Report · Acceptance letter]

23 Mar 2022

Dear Prof. Wilson,

We are delighted to inform you that your manuscript, "A novel CSP C-terminal epitope targeted by an antibody with protective activity against </i>Plasmodium falciparum</i>.," has been formally accepted for publication in PLOS Pathogens.

Best regards,

Kasturi Haldar

Editor-in-Chief

PLOS Pathogens

orcid.org/0000-0001-5065-158X

Michael Malim

Editor-in-Chief

PLOS Pathogens

orcid.org/0000-0002-7699-2064